# Transcriptome-Based Analysis of the Response Mechanism of Leopard Coralgrouper Liver at Different Flow Velocities

**Min Yang** [1,2]**, Jin Gao** [3,4,5]**, Hongji Ke** [3]**, Yongbo Wang** [3,4,5]**, Jinye Liu** [3,4]**, Xin Wen** [2]**, Shuyuan Fu** [3,4,5] **and Jiang Wang** [1,*]

1 College of Science, Hainan University, Haikou 570228, China
2 College of Oceanography, Hainan University, Haikou 570228, China
3 Hainan Provincial Key Laboratory of Tropical Maricultural Technologies, Hainan Provincial Engineering Research Center for Tropical Sea-Farming, Hainan Academy of Ocean and Fisheries Sciences, Haikou 571126, China
4 Key Laboratory of Utilization and Conservation for Tropical Marine Bioresources, Ministry of Education, Sanya 572022, China
5 Yazhou Bay Innovation Institute, Hainan Tropical Ocean University, Sanya 572025, China
* Correspondence: wamgjgamg9623@126.com

**Abstract:** The velocity of water can cause differences in the growth of inhabitant fish, thereby functioning as an important aquatic environmental factor. In order to explore the response mechanism of Leopard Coralgrouper (*Plectropomus leopardus*) under different flow velocities, *P. leopardus* fish of the same sizes were fed in water characterized by three different flow rates (5 cm/s, 10 cm/s, and 20 cm/s) for 150 d, after which the differences in growth and the levels of liver oxidative stress kinase were analyzed. Additionally, potential underlying regulatory pathways were explored using comparative transcriptomic approaches. The results showed that in the feeding environment involving a flow rate of 10 cm/s, the weight increase of *P. leopardus* and its liver contents of glutathione peroxidase (GPX), alanine aminotransferase (ALT), and superoxide dismutase (T-SOD) were significantly higher than for the other two groups. After matching the comparative transcriptomic results between group pairs (5 cm/s and 10 cm/s, 5 cm/s and 20 cm/s, and 10 cm/s and 20 cm/s), a total of 225 common differential gene expressions were screened. The KEGG pathway enrichment analysis showed that these genes were mainly involved in protein processing in the endoplasmic reticulum, the TGF-β signaling pathway, fatty acid metabolism, fatty acid synthesis, and other pathways. These results not only provide a theoretical basis for the selection of flow velocity in the culture environment of *P. leopardus,* but also reveal its potential means of adaption to different flow velocities.

**Keywords:** flow velocities; growth; *Plectropomus leopardus*; transcriptomics

## 1. Introduction

*Plectropomus leopardus* is a rare, high-grade, edible, and ornamental fish. At present, existing research on *P. leopardus* has primarily focused on conservation ecology [1], its reproduction [2], its mitochondrial genome [3], and its growth [4], as well as its response to environmental factors such as light wavelengths and the effect of food on its body coloring [5,6]. These findings suggest that the environment alters its growth from multiple avenues. In modern times, *P. leopardus* is one of the most common species of grouper in factory farming, with an annual output of nearly 10,000 tons and an output value of more than 2.6 billion units, thereby serving as a species that confers significant economic and social benefits. There are two breeding methods employed for *P. leopardus*. The first is factory-flow aquaculture, and this is the most important breeding mode, which features the advantages of convenient operation, a stable water environment, and disease prevention and control, among others [7]. The second is cage culture, which features the advantages of fast growth, largeness of scale, high efficiency, lower rates of disease, convenience for

intensive and industrialized production, and many other advantages [8]. In recent years, China has gradually shifted towards utilizing the second breeding mode of *P. leopardus*. The physical and chemical conditions that characterize aquaculture water comprise the main difference between the two models, especially the flow velocity, and these cause differences in the growth of the contained fish [9]. A previous study showed that there were significant differences in the liver fat content of *P. leopardus* at different flow velocities [10]. However, little is known about the growth differences of *P. leopardus* at different flow velocities and how these growth differences are related to the regulation of gene expression in the liver.

Under the influence of flow velocity, migratory and reef-dwelling fish in nature must choose a suitable habitat in order to balance the energy consumed by their growth and development against that required for environmental adaptation [9]. Therefore, flow velocity (referring to the total volume of fluid flowing per unit of time past a cross-section of a channel) has been widely studied as an important environmental factor in fish farming environments. For example, the behavior and metabolism of juvenile hybrid sturgeon can change at different flow velocities; as the flow rate increases, the frequency of tail wagging and the rate of oxygen consumption correspondingly increases [11]. Different flow rates exert significant effects on the body fat content of *Paralichthys olivaceus* juveniles [12]. Similarly, *Paralichthys californicus* is affected by changes in flow velocity, which can affect the efficiency of the feed conversion and utilization of juveniles, in turn directly affecting their growth performance [13]. In a recirculating aquaculture system, increases in flow velocity reduce the accumulation of total ammonia nitrogen and total vibrio in the water body, thereby improving the aquaculture environment and increasing the growth rate of *Scophthalmus maximus* juveniles [14]. Similarly, with the increasing flow velocity, the final body weight, body length, growth efficiency, net weight gain, daily growth rate, and daily gain of juveniles of *Acipenser baerii* have all been found to gradually increase, while the feed coefficient was found to gradually decrease [15]. As an important organ in metabolism, the liver is closely related to growth. Changes in the physiological indicators of the liver can effectively reflect the current state of the individual. Such indicators include superoxide dismutase (SOD), glutamic pyruvic transaminase (ALT), and glutathione peroxidase (GPX) [16]. A previous study found that flow velocity can affect the growth rate and physiological state of *Micropterus salmoides* juveniles in captivity [17]. Therefore, studying the effect of flow velocity on aquatic organisms can help to improve the adjustment of the physical and chemical conditions of aquaculture conditions, thereby improving the economic benefits conferred by tailored aquaculture.

The collection of transcriptomic data is an important tool employed for the large-scale screening of genes, as it can clearly display the transient expression rules of genes in specific tissues [18]. RNA-seq is high-throughput sequencing technology used for sequencing analysis that reflects the expression levels of mRNA, small RNA, noncoding RNA, etc. Over the past decade, RNA-Seq technology has rapidly developed to become an indispensable tool for analyzing differential gene expression at the transcriptome level [19]. RNA-Seq can detect whole transcripts, which is of great significance in basic biological research, its sensitivity is relatively high, and it has a wide range of applications [20,21]. The transcription of genes is regulated not only by the genetic differences of different species, but also externally affected by factors such as the environment, food, and human activities, so transcriptomics is widely used to reveal the growth and development of organisms, their physiological responses to stress, their disease resistance, and other biological processes [22,23]. In recent years, transcriptomic technology has been widely employed in the study of aquatic animal physiology and biochemistry [24,25]. For example, in analyzing the differentially expressed gene ion channels of *Nibea albiflora* muscle under different concentrations of salinity stress, it has been found that the adaptive mechanism employed by the fish to changes in its environment is regulated by multiple signaling pathways, whereby changes in salinity affect the function of its immune system, and the results obtained by this study have provided a reference for the ideal salinity of the aquaculture water used to habituate *Nibea albiflora* [26]. A transcriptomic analysis conducted on red and white colored *Cyprinus carpio* skin yielded

52,902 SNP loci suitable for carp, and these can be used to assist in the selection and breeding of carp with specific body colors [27]. A transcriptomic analysis of *Pelteobagrus fulvidraco* individuals at low and normal temperatures was used to explore their protein and fat digestion and absorption signaling pathways, and to identify the key genes related to their resistance to low-temperature stress [28]. After adding mannose oligosaccharide to the feed, the immune-related pathways of the intestinal flora of *E. fuscoguttatus* × *E. lanceolatus* were investigated, and the results now provide a basis for the disease resistance mechanisms employed by grouper [29]. These studies show that transcriptomics is an indispensable tool to analyze the adaptation of aquatic animals to different environments, as it can be used to reveal the mechanisms of biological regulation and the accompanying phenotypic traits at the mRNA level. This paper firstly compared the growth performance of *P. leopardus* at three different flow rates. Subsequently, based on obtained RNA-Seq data, the gene expression differences underlying the different growth performances were analyzed to provide basic data for an in-depth study of the physicochemical conditions of artificially-cultured *P. leopardus*.

## 2. Materials and Methods

### 2.1. Experimental Materials

The experimental materials were obtained from the Hainan Seawater Breeding Center (Yelin Village, Qionghai City). Based on previous research results [10], this study used flow rates of 5 cm/s, 10 cm/s, and 20 cm/s as the differential conditions for the breeding experiment. In the same breeding batch and rearing environment, individuals with a body length of $20 \pm 2$ cm and an average weight of $100 \pm 5$ g were selected (As shown in Table 1). A total of 1500 (500 fish in each of the three groups) juveniles were placed conditions of 20 cm/s (SP20), 10 cm/s (SP10), and 5 cm/s (SP5), and were cultured in experimental water tanks. The artificial diet (formulated diet of grouper) used in the experiment was from Fujian Tianma Science and Technology Group Co., Ltd. (Fuqing, China), and the feeding amount was 4% of the fish body weight, that is, 2.5 g/tail, three times a day. Every 30 days, 100 individuals in each experimental group were randomly selected to measure their weight and body length after MS-222 anesthesia, and the average value was calculated; then, they were quickly transferred back to the experimental water tanks. After 5 months, experimental fish ($n = 3$) with similar characteristics and good vitality were selected from each group (total of nine individuals), and their liver tissue samples were obtained by operation conducted on ice. The obtained samples were temporarily stored in liquid nitrogen before being transferred to $-80$ °C for storage. The samples were subsequently tested for their enzyme activity and transcriptome analysis.

**Table 1.** Changes in body length and body weight of *P. leopardus* under different flow velocities.

| | 20 cm/s | | 10 cm/s | | 5 cm/s | |
|---|---|---|---|---|---|---|
| | Average Body Length (cm) | Average Weight (g) | Average Body Length(cm) | Average Weight (g) | Average Body Length(cm) | Average Weight (g) |
| Month 1 | $19.30 \pm 1.3$ | $104.18 \pm 2.6$ | $19.47 \pm 1.2$ | $105.22 \pm 2.3$ | $19.40 \pm 0.9$ | $104.16 \pm 2.3$ |
| Month 2 | $22.96 \pm 1.3$ | $174.57 \pm 2.4$ | $22.65 \pm 1.1$ | $179.38 \pm 2.8$ | $22.75 \pm 1.1$ | $166.41 \pm 3.6$ |
| Month 3 | $24.81 \pm 1.4$ | $227.92 \pm 3.7$ | $24.35 \pm 0.9$ | $226.38 \pm 3.2$ | $23.95 \pm 1.2$ | $225.05 \pm 3.9$ |
| Month 4 | $26.47 \pm 1.4$ | $263.68 \pm 3.9$ | $25.82 \pm 1.2$ | $288.75 \pm 2.4$ | $26.03 \pm 0.8$ | $262.07 \pm 3.1$ |
| Month 5 | $27.83 \pm 1.5$ | $344.25 \pm 4.6$ | $28.03 \pm 1.1$ | $358.97 \pm 3.3$ | $27.60 \pm 1.3$ | $330.73 \pm 4.9$ |

### 2.2. Enzyme Activity Analysis

Amounts of 100–250 mg of liver tissue were weighed and then rinsed with 0.85% saline (pre-cooled) 2–3 times before removing residual blood in the tissue, drying the water droplets on the surface of the tissue with filter paper, recording the weights of the samples, and placing them into respective centrifuge tubes. A corresponding volume of normal saline was pipetted (1 g to 9 mL), and then the tissue was rapidly chopped in an ice-water



bath and ground with a homogenizer for subsequent use. The resulting tissue homogenate (10%) was then centrifuged (12,000 rpm, 10 min, 4 °C), and the resulting supernatant was added with 0.85% normal saline at a ratio of 1:9 to prepare a 1% homogenate sample for the detection of the enzyme activity.

The enzyme activity detection kits were purchased from Nanjing Jiancheng Biological Company, and the detection object was liver tissue. Detailed experimental steps can be obtained by referencing the kit instructions. Glutathione peroxidase (GPX, H545−1), alanine aminotransferase (ALT, C009-2), and superoxide dismutase (T-SOD, A001-1) were detected. All of the results were measured using an automatic biochemical analyzer for absorbance at a specific wavelength, and the corresponding concentrations were calculated using the formula or standard curve supplied in the manual.

### 2.3. RNA Isolation and Transcriptome Sequencing

The total RNA extraction, quality detection, and concentration detection of the liver samples obtained from the nine samples (5 cm/s, 10 cm/s, and 20 cm/s; three fishes per group) were performed using the Trizol Kit (Invitrogen, Carlsbad, CA, USA), Bioanalyzer 2100, and RNA 6000 Nano LabChip Kit (Agilent, CA, USA), respectively.

First, the Illumina TruSeq RNA sample Preparation Kit (Illumina, San Diego, CA, USA) was used to prepare the samples for the subsequent library construction, and then the quality-ensured total RNA was enriched for eukaryotic mRNA using magnetic beads attached to Oligo (dT) to build a cDNA library from 5 μg of total RNA taken from each group of samples. A fragmentation reagent (Fragmentation Buffer) was used to randomly break the extracted mRNA into short fragments, and these were used as templates to synthesize the strands of cDNA incorporating random hexamers. Then, two-strand synthesis was carried out with buffer, dNTPs, and RNaseH under the combined action of DNA Polymerase I. AMPure XP beads were used to purify the double-stranded product. The activities of two different polymerases (T4 DNA polymerase and Klenow DNA polymerase) were utilized to repair the sticky ends of DNA into blunt ends, and the base A and adapters were added at the 3′ ends. The fragments targeted by the AMPureXp beads were amplified by PCR to obtain the final sequencing library, and the qualified library was sequenced using the paired-end sequencing method.

### 2.4. Data Assembly, Annotation, and Variance Analysis

We filtered the machine data (raw data) and removed low-quality reads to obtain high-quality data (clean data) for the subsequent analysis. Using HISAT2 software to map the high-quality data to for the reference genome of *P. leopardus* (https://db.cngb.org/search/project/CNP0000859, 5 July 2021), the alignment results were assembled using StringTie software to obtain transcripts, and based on transcription, a follow-up analysis was carried out. The gene function annotations of the transcripts were based on the reference genome, unannotated fragments were regarded as new transgenes of the species, the new genes were aligned in the public database using BLAST software, their amino acid sequences were predicted, and then corresponding proteins were identified using the Pfam database. The database was annotated using HMMER software functions. DEGseq was used to analyze the differential expression of genes in all experimental samples. The $p$ value was used to adjust the $p$ value of the experimental results. Significant difference intervals were defined as $p$-values < 0.05 and |log2 (fold change)| > 1. Subsequently, Gene Ontology and KEGG pathway enrichment analysis was performed on the screened differentially expressed genes. Metascape software was used for Gene Ontology and KEGG pathway enrichment analysis. Based on the Wallenius noncentral hyper-geometric mathematical model of the R language GOseq package, GO enrichment analysis was performed on the screened differentially expressed genes. Using the Pathway in the KEGG database as a unit, the hypergeometric test method of KOBAS 2.0 software was used to detect the significantly enriched pathways in the differentially expressed genes.

## 2.5. Quantitative Real-Time PCR (qRT-PCR)

The reverse-transcribed cDNA of *P. leopardus* liver tissue was used as the template for the real-time fluorescence quantitative PCR, and each step was performed according to the experimental steps outlined by the kit. The reaction system was comprised of 10 μL in total in descending order of the addition of the following: 2× Power Green qPCR Mix (50%) (Dongsheng Biotechnology, Guangzhou, China), ddH2O (41%), cDNA (5%), and upstream and downstream primers (10 μmol·L$^{-1}$) (2%). The primer information is listed in Table 2. The program used was as follows: pre-denaturation at 95 °C for 30 s, 95 °C for 5 s, 55 °C for 30 s, and 72 °C for 30 s (for a total of 40 cycles); a dissolution curve was drawn using 95 °C for 5 s, 65 °C for 60 s, and cooling at 50 °C for 30 s. The samples each underwent qRT-PCR three times. In order to reduce the error of the experimental operation and the results, the real-time quantitative PCR results were calculated according to the formula $2^{-\Delta\Delta Ct}$.

**Table 2.** Sequences of primers used in this study.

| Gene Name | Forward Sequences | Reverse Sequences |
|---|---|---|
| *cyp7b1* | ACTTCATCGCCCTCTACCCTC | TGAGCCTCTGACCGTCTTTG |
| *cpt1a* | AGCACCTGACTGACCGTAAGC | GCATCTCAAGTTCACTGGGTAAG |
| *irs2* | TGACATCAGCGACCCTTGTG | CGCCACTACTCTCTGTTGACG |
| *Kmt5c* | GCAGCAAAGACTGGAGCAAG | TCGGTGAACTCATCTGGCAC |
| *acod* | AGCAATGTTCTCCCTGAGGC | CCAAAGCAAGGTCAAAGGATG |
| *cyp2j2* | GGCAACTTATTCTCTGTGGATTTC | GCTGTCTCCCTGATTTACCAGTG |
| *acsbg2* | GCAGCAGAAGAGCCTGACCTAC | TAGATGCCAACAGCAAACCC |
| *β-actin* | CACCACAGCCGAGAGGGA | TCTGGGCAACGGAACCTCT |

## 2.6. Data Processing

SPSS 22.0 was used for the statistical analysis of the data in this experiment, and the data of each group were presented as the mean ± standard deviation (mean ± SD). The two-tailed t-test was used to measure the difference in gene expression detected by real-time quantitative PCR. A *p*-value < 0.05 was regarded to indicate a significant difference, and this has been denoted with "*".

## 3. Results and Analysis

### 3.1. Growth Analysis

As shown in Table 1, there was no significant difference found in the body lengths of the fish under the three flow velocities (the average difference was less than 1 cm). However, there was a significant difference found in their body weights, among which, the 10 cm/s flow rate condition produced the greatest weight gain, which was significantly higher than that of the other two groups in the fifth month, thereby indicating that 10 cm/s could be used as a reference flow rate for *P. leopardus* culture.

### 3.2. Enzyme Activity Analysis

The liver tissue enzyme (Table 3) biopsy results showed that the levels of alanine aminotransferase (ALT), superoxide dismutase (SOD), and glutathione peroxidase (GPX) in the 10 cm/s test group were significantly higher than other two groups. Compared with the 5 cm/s group, the values of these three enzymes in the 20 cm/s condition more closely resembled those measured in the 10 cm/s test group.

**Table 3.** Comparison of the liver enzyme activities of *P. leopardus* under different flow velocities.

| Enzyme Activity (U/g) | Flow Velocities | | |
|---|---|---|---|
| | **20 cm/s** | **10 cm/s** | **5 cm/s** |
| alanine aminotransferase(ALT) | 9.62 [b] | 12.17 [a] | 10.72 [b] |
| superoxide dismutase(SOD) | 560.30 [b] | 586.02 [a] | 444.79 [c] |
| glutathione peroxidase(GPX) | 48.28 [b] | 62.10 [a] | 5.70 [c] |

"a, b, c" mean that the enzyme has significant difference (*p < 0.05*).

### 3.3. Transcriptome Sequencing

In this study, transcriptome sequencing (Illumina) was performed on the livers of *P. leopardus* under the flow rate conditions of 20 cm/s, 10 cm/s, and 5 cm/s. The data output statistics of each sample are shown in Table 4. After filtering the low-quality data, a total of 75.79 Gb of clean data were obtained. The Hista was used to match these data to the genome, as shown in Table 4. The comparison efficiency between the reads of each sample and the reference genome was between 92.63% and 93.78%, and the quality of the data met the requirements for the subsequent analysis.

**Table 4.** Quality of transcriptome sequencing data.

| Sample | Clean Reads | Clean Bases | Q30 (%) | GC Content (%) | Total Mapped |
|--------|-------------|-------------|---------|----------------|--------------|
| SP20-1 | 32,524,608 | 9.06 G | 95.11 | 51.68 | 30,229,755 (92.94%) |
| SP20-2 | 24,959,181 | 6.95 G | 95.16 | 50.72 | 23,119,553 (92.63%) |
| SP20-3 | 29,608,098 | 8.26 G | 94.88 | 51.56 | 27,452,439 (92.72%) |
| SP10-1 | 26,861,072 | 7.50 G | 95.30 | 51.42 | 25,190,298 (93.78%) |
| SP10-2 | 35,829,110 | 9.99 G | 95.05 | 51.51 | 33,509,319 (93.53%) |
| SP10-3 | 23,503,733 | 6.56 G | 94.98 | 50.44 | 22,011,276 (93.65%) |
| SP5-1 | 27,742,612 | 7.74 G | 95.33 | 50.71 | 25,838,704 (93.14%) |
| SP5-2 | 30,041,710 | 8.38 G | 95.12 | 51.02 | 27,966,268 (93.09%) |
| SP5-3 | 21,989,239 | 6.14 G | 95.44 | 50.82 | 20,452,207 (93.01%) |

### 3.4. Differentially Expressed Gene Analysis

Pairwise comparisons were made between the different flow velocity groups to screen for differential genes. Subsequently, these differences were focused on, common genes were further screened, and Venn diagrams were drawn. As shown in Figure 1. The number of unique and common differential genes was compared pairwise, and 225 genes were found to belong to the features that were common between them.

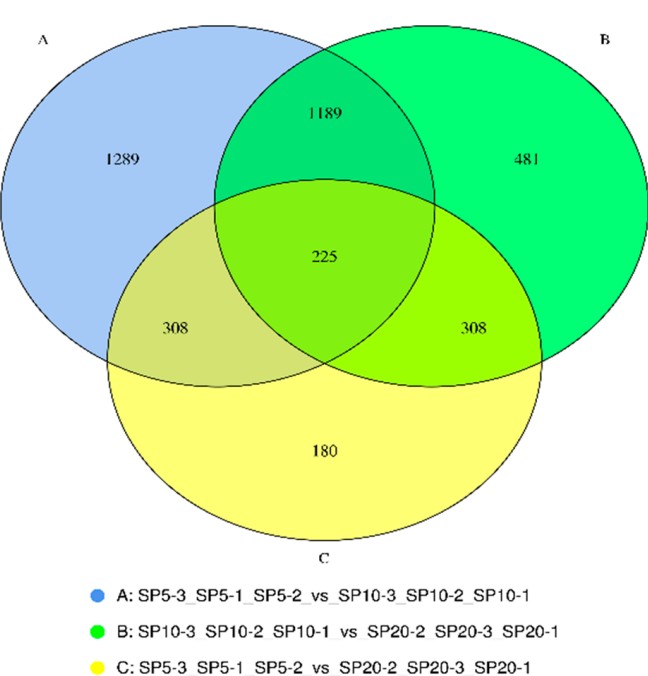

A: SP5-3_SP5-1_SP5-2 vs SP10-3_SP10-2_SP10-1
B: SP10-3 SP10-2 SP10-1 vs SP20-2 SP20-3 SP20-1
C: SP5-3 SP5-1 SP5-2 vs SP20-2 SP20-3 SP20-1

**Figure 1.** The number of differential genes between different groups (SP5: flow velocity 5 cm/s; SP10: flow velocity 10 cm/s; SP20: flow velocity 20 cm/s; "−1, −2, −3" indicates three parallel samples of the same experimental group).

### 3.5. Quantitative Real-Time PCR (qRT-PCR)

qRT-PCR validation can be used to effectively judge the reliability of the differential transcriptomic data. As shown in Table 5, among the seven selected genes (four downregulated, three upregulated; primers derived from previous studies [1]), the data obtained from transcriptome and qRT-PCR analysis showed consistent trends (the difference between 10 cm/s and 20 cm/s), thereby indicating that the sequencing results were both reliable and reproducible.

**Table 5.** Analysis of differentially expressed genes.

| Gene Name | Gene ID | log2fold Change | log2fold Change |
|---|---|---|---|
| *cyp7b1* | utg000043l-1.624 | −2.87 | −1.96 * |
| *cpt1a* | utg000134l-0.116 | −3.66 | −1.87 * |
| *irs2* | utg000150l-1.275 | −4.32 | −2.35 * |
| *kmt5c* | utg000519l-0.406 | −2.35 | −2.89 * |
| *cyp2j2* | utg000129l-0.32 | 3.96 | 5.22 * |
| *acsbg2* | utg000537l-0.141 | 4.31 | 3.87 * |
| *acod* | utg000003l-2.243 | 3.11 | 4.21 * |

"*" means that the gene has significant difference ($p < 0.05$).

### 3.6. GO and KEGG Analysis

GO (Figure 2) enrichment analysis ($p$-value $< 0.05$) was performed on the 225 focused genes. The resulting GO clusters were mainly divided into two categories: (1) biological processes and (2) molecular functions. In the molecular function category, all were found to be related to the activity of enzymes, among which "pyrophosphatase activity" and "hydrolase activity, acting on acid anhydrides" were particularly prominent. In addition, "protein refolding" and "alcohol metabolic process" were relevant biological processes.

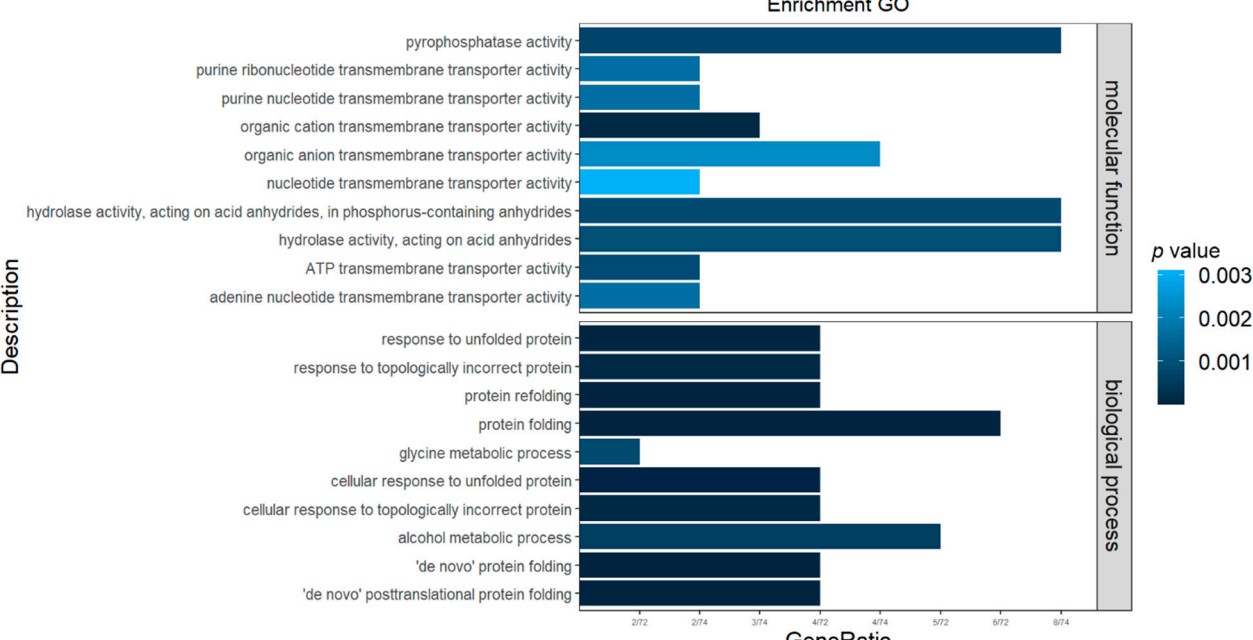

**Figure 2.** Go enrichment analysis.

As shown in Figure 3, 10 pathways were significantly enriched ($p$-value $< 0.05$) by the KEGG analysis, for example, protein processing in the endoplasmic reticulum, TGF-beta signaling pathway, fatty acid metabolism, fatty acid biosynthesis, etc.

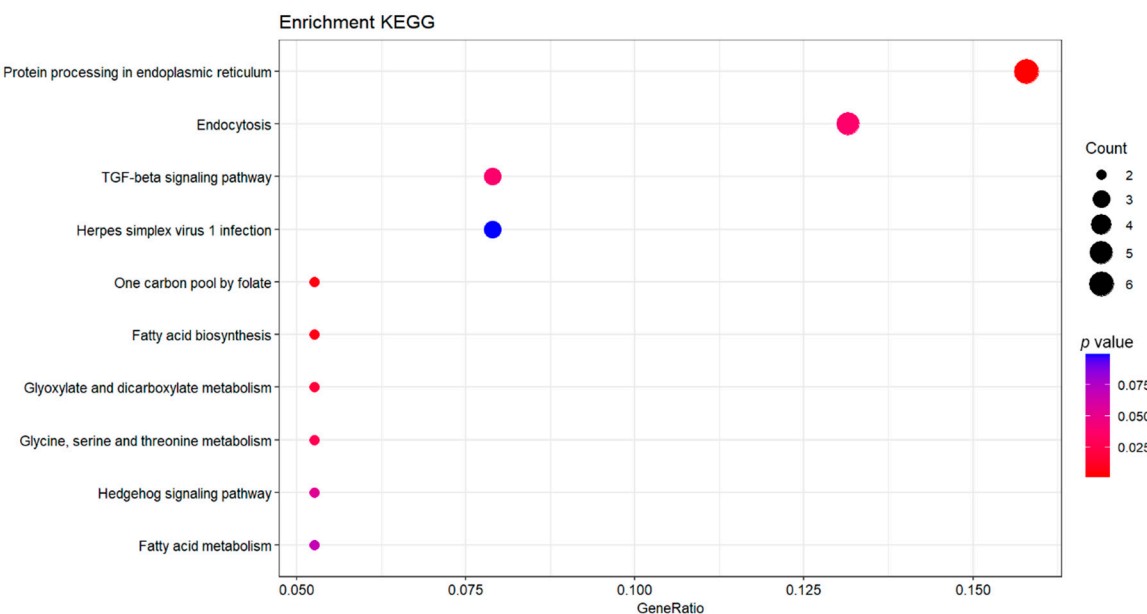

**Figure 3.** KEGG enrichment analysis.

## 4. Discussion

### 4.1. The Effect of Flow Velocity on Fish Growth

Water velocity is an important physicochemical factor that regulates the process of fish growth and development. It can change the frequency of fish swimming, enhance metabolic activity, and regulate its physiological functions, thereby affecting nutrient intake, growth, and development [13]. A previous study found that the weight gain rate and specific growth rate of *Micropterus salmoides* at a flow rate of 40 cm/s were significantly higher than those in static water, and in 20 cm/s and 60 cm/s flow rate environments [30]. These findings are consistent with the results obtained by the current study. After 5 months of culture in a hydrological environment with a flow rate of 10 cm/s, compared with 5 cm/s and 20 cm/s, the body weight of *P. leopardus* increased significantly. Similar results were also found in studies on *Epinephelus coioides* [31] and *Morone saxatilis* [32]. These results confirm that the appropriate flow rate is of great significance for the growth and development of fish. Under certain flow velocities, fish may enhance their spontaneous activity, while at high flow rates, fish must consume more energy to maintain their swimming velocity to counteract the flow rate, and correspondingly reduce the energy allocated to growth [33]. In addition, a stable and suitable water flow velocity can not only benefit metabolism, but also reduce the attachment of water parasites, thereby minimizing damage to the fish body as well as the need for artificial parasite treatments and the corresponding consumption of such chemicals. Water velocity therefore represents a low-cost, sustainable, and ecological means through which to avoid the negative consequences of ectoparasites. At present, the flow rate of *P. leopardus* in factory flow aquaculture is generally maintained at approximately 10 cm/s, while the average flow rate of the sea cage culture is about 40 cm/s. Interestingly, a previous study found that at flow rates of 10 cm/s and 40 cm/s [10], the area and density of hepatic lipid droplets were significantly higher in the low-flow group than in the high-flow group. This also shows that excessive flow rates require fish to engage in higher energy consumption to maintain balance, thereby impacting growth [34]. To explore the underlying mechanisms of this process, many studies have investigated the relevant enzymes in the liver tissue based on ambient flow velocities.

### 4.2. Effects of Various Flow Velocities on P. leopardus

Under different environments, teleost fish adjust their corresponding survival strategies through adaptation. Such adaptations include improving the lipid metabolism, changing the balance of unsaturated fatty acids, and reducing the frequency of activities to

maintain the loss of body heat, among others. Such adjustments made to adapt to the environment comprise the purpose of physiology and behavioral activities [35]. As the main site of metabolism, the liver can accurately reflect the physiological status of fish during exercise [36]. Alanine aminotransferase (ALT), glutathione peroxidase (GPX), and superoxide dismutase (T-SOD) jointly represent the physiological levels of liver antioxidants in response to any given environment. A previous study found that the serum levels of SOD in *Oncorhynchus keta* [37] were significantly increased after 11 weeks of survival in an environment with flow rates of 2 cm/s, 13 cm/s, and 22 cm/s; the SOD levels and T-AOC (total antioxidant capacity) of *Barbodes schwanenfeldi* were also found to be increased with increasing water flow velocities [38]. These results show that different flow velocities change the antioxidant level of fish. The current study found that the levels of enzymes (SOD, ALT, and GPX) in the 10 cm/s flow rate group were significantly higher than those in the other two groups. Both GPX and SOD are ubiquitous throughout the body, and they can remove peroxide and hydroxyl radicals that are generated by the process of cellular respiration and metabolism, thereby performing an antioxidant role and protecting the structure and function of cell membranes from interference and damage. Interestingly, in the 10 cm/s flow rate group, the levels of alanine aminotransferase were determined to be relatively high, thereby indicating that the liver body may have possessed lesions, which may explain the relatively high detected levels of antioxidant enzymes (GPX and SOD) to protect the body. In the 10 cm/s flow rate condition, *P. leopardus* altered its metabolic balance of enzymatic activity. Although the results of the present study cannot determine whether such a balance of oxidative stress is beneficial or detrimental to the physiology of *P. leopardus*, combined with the differences in body weight gain, the results indicate that the observed oxidative stress level is permitting of growth, and that 10 cm/s appears to represent a water flow rate condition that is suitable for the breeding of *P. leopardus*. Similarly, after rearing for 154 days under low flow rate (0.2 m/s), medium flow rate (0.4 m/s), and high flow rate (0.6 m/s), the liver SOD activity of the medium flow rate group was measured to be the highest. Moreover, high flow velocities can cause damage to hepatocytes, thereby resulting in the severe vacuolation and deviated aggregation of the nuclei [31]. These results indicate that the appropriate water flow rate can enhance the antioxidant capacity of fish.

### 4.3. Regulatory Pathway

Adaptive changes employed by the body must be regulated by gene expression to affect the synthesis and metabolism of proteins. In this study, RNA-Seq was used to screen differentially expressed genes under different flow rate conditions, and the focused genes (Figure 1) were then enriched through KEGG pathway analysis, thereby significantly enriching major regulatory pathways that are predicted to be involved in responding to different flow rates (Figure 3). Among them, endoplasmic reticulum protein processing was particularly prominent, indicating that changes in the flow rate alter the metabolic synthesis of hepatic proteins in *P. leopardus* to affect its metabolism. This is consistent with previous findings that revealed water flow velocity exerted an effect on protein synthesis in fish, and that inconsistent movement frequencies in response to different flow rates alter the liver protein deposition in *Barbodes schwanenfeldi* [39]. The liver is an important organ for the synthesis of digestive enzymes. In *Micropterus salmoides* subjected to different flow rates, changes in liver function affect the levels of lipase, protease, amylase, and other digestive enzymes, as well as glutamate dehydrogenase, hormone-sensitive lipase, and acetone metabolic activity, as reflected by acid kinase [30], which is a potential cause for the difference in growth. This also indirectly indicates that the difference in growth observed in *P. leopardus* at the three tested flow rates is closely related to the processing of ER proteins in the liver, because these enzymes affect the metabolic efficiency of the body. Correspondingly, fatty acid oxidation is primarily carried out in the liver, thereby allocating the energy required for bodily growth, development, and material metabolism [40]. Previous studies have found that water flow greatly affects the fatty acid content and composition of muscles

in *Barbodes schwanenfeldi* [38]. In this study, both fatty acid metabolism and related fatty acid synthesis pathways at 10 cm/s were found to be significantly enriched, indicating that the balance between fatty acid metabolism and synthesis differs in water bodies characterized by different flow rates. Similarly, serum fatty acids in *Micropterus salmoides* were found to decrease with increasing flow rates, and this was more pronounced at higher flow rates [30]. This may be because the body requires more energy to maintain homeostasis to cope under a high flow rate environment, and fatty acids can be oxidized and decomposed into $CO_2$ and $H_2O$ under a sufficient supply of oxygen, thereby releasing a large amount of energy that the body can utilize. On the contrary, the flow rate was found to be closely related to the fatty acid composition and growth of *Rhynchocypris lagowskii* juveniles [34], whereby the content of unsaturated fatty acids measured in the high flow rate group was significantly higher than that in the control group, which was a similar result to that obtained by a study on *Cyprinus carpio* [41]. Although these results suggest that the strategies adopted by different fish to respond to different flow rates are not consistent, flow rate factors have consistently been found to alter the fatty acid metabolism of various fish. Therefore, we speculate that at a flow rate of 10 cm/s, the growth performance is superior and the antioxidant capacity is higher. The potential regulation pathway underlying this outcome is that the body adjusts its metabolism and synthesis of liver proteins, thereby affecting the metabolism of fatty acids. The results of this study not only provide new data on the regulatory mechanism underlying the flow rate adaptability of *P. leopardus*, but also lay a theoretical foundation for ideal flow rate conditions in the context of factory farming and the selection of aquaculture sea areas.

## 5. Conclusions

This paper preliminarily discussed the changes in the expression of related genes at the transcriptional level among individuals of *P. leopardus* under various flow velocities. The underlying adaptive mechanisms were also investigated. This study not only provides a theoretical basis for the analysis of molecular genetic mechanisms underlying the economic traits of *P. leopardus* and the selection of new varieties, but also lays a foundation for the further development of molecular markers, selection of improved varieties, and selection of aquaculture breeding areas.

**Author Contributions:** M.Y. and J.W. are responsible for the design, implementation, and writing of this experiment; X.W., H.K. and J.L. are responsible for the daily maintenance of leopard-skinned seabass breeding; S.F., J.G. and Y.W. are responsible for data collection and analysis. All authors have read and agreed to the published version of the manuscript.

**Funding:** This work was supported by Key R&D Project in Hainan (ZDYF2020093); the Hainan Provincial Natural Science Foundation of China (320QN208); the initial fund from Hainan University for R&D KYQD (ZR) 20012.

**Institutional Review Board Statement:** The Ethics Committee of Experimental Animals at Hainan University approved this study (HNUAUCC-2022-00038).

**Data Availability Statement:** Not applicable.

**Conflicts of Interest:** The authors declare no conflict of interest.

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
