# Peer review of "Transcriptome-Based Analysis of the Response Mechanism of Leopard Coralgrouper Liver at Different Flow Velocities"

_fishes, doi:10.3390/fishes7050279_

Round 1

Reviewer 1 Report

Review of

Transcriptome-based Analysis of the Response Mechanism

This is an interesting project aiming to elucidate the gene expression responsible for increased growth in fish raised at an optimal flow speed.  The work has promise but the writing makes it difficult to interpret.  There are significant deficiencies in the methods that prevent adequate evaluation of the results.

Introduction

The language of this section needs work to make sure the authors use appropriate sentence structure.

Line 50-52 are a critical concept to the study and should be expanded for clarity.

The explanation of RNA-Seq is weak.  Language such as a “possessing the ability of whole transcripts” is not clear. The sentence on lines 75-78 is grammatically incorrect.

No hypothesis is given.

Methods

The methods section alternates between various styles, from instructions (e.g lines 118-122) to present-tense passive voice (e.g. lines 160-161) to past-tense passive voice (e.g. lines 122-125).  Sentence by sentence it needs to be rewritten to carefully to maintain one voice. Some things are simply not sentences (e.g. line 128-129), and other sentences are run-on sentences (lines 110-113).

How were the three fish from each condition selected?  The MS says they had the “same specificiation and good vitality”.  Were they randomly selected from the 1500?

For the fish size data in Table 1, how many fish were sampled?  Were they measured and returned to the tank or were they sacrificed.  I may have missed it, but sample size and method of sampling needs to be made clear.

The methods used to measure enzymatic activity needs greater depth.  What kits were they?  What device was used to measure absorption?  Did the authors do a standard curve or rely upon something in the kit instructions?  If they did not do an independent standard curve, the reliability of the data is suspect. 

There appears to be a missing table.  Methods refers to the primers in Table 1, but Table 1 is the results for fish size.

No statistical analysis information is given.

Results

For table 1, results are described, but the full results of the statistical analysis are not given. The text simply refers to body mass as being significantly greater in the 10cm/s group. At all time points?  Just at the end?  Are any of these fish the same fish with repeat measures?  The sampling information needed in the methods would make that clear.

The absence of primer information contributes to the weak presentation of Table 4.  Why these genes? 

Discussion

This section needs to be completely rewritten.  Start with a summary of your results.  Beginning largemouth bass misdirects the reader.  Focus on P. leopardus first.  The 2nd paragraph indicates it i

Author Response

Dear editor: 
The resubmitted manuscript has been performed critical changes recommended by the referees.
Many thanks for your positive comments to our manuscript. It is great pleasure for us to cooperate with you to obtain a publishable manuscript. We're terribly sorry for our poor writing proficiency which brought a lot of language logic errors. We have carefully revised the manuscript according to reviewer’s comments and all modifications were highlighted in red.
The authors declare no conflict of interests regarding the publication of this article.
We really appreciate your consideration of our manuscript.

Sincerely yours,    
Min Yang.

Reviewer 2 Report

Abstract:

it is not clear how many experimental groups there are: first we talk about 4 experimental groups and then for the transcriptomic analysis we indicate 2 experimental groups, which ones have been selected? this information should already be given in the abstract

M&M section:

provide more detailed information on farming conditions: type of feed, amount of feed administered daily (ad libitum?). it is essential to have these indications when evaluating the differences between growth rates

line134: the Authors referred to 9 groups. I strongly suggest to better define and describe the experimental groups

164: indicate the software used for Gene Ontology and KEGG pathway enrichment analysis

line 167: The Authors referred to “leopard gill bass tissue” instead liver samples.

Statistical analysis: lines 180-183: the authors did not refer to which statistical test was applied for the growth and enzyme activity data. Pleas add and add the letters or asterisks for statistical significance in table 1

In table 2 please add standard deviation or SEM

Lines 212-214: please clarify the characteristics of the 225. Are they differentially expressed in all the pair-wise comparisons? From the Venn Diagram it seems that the 225 genes are not differentially expressed but on the contrary they are common among all the experimental groups. Please clarify

Line 214: probably there is a refuse

Regarding real time PCR there are several points unclear:

Which are the reference genes used for quantification?

Table 4 is not clear: the statistical differences here reported are referred to which comparison? Which are the experimental groups involved in this analysis? I strongly suggest realizing a graph with histograms comparing the log2 fold change of target genes among different experimental groups.

GO and KEGG analysis: is not clear why the Authors performed such analyses on 225 genes. Which is the meaning? I strongly suggest to perform such analyses on DEG for each pair-wise.

Please revise the discussion section taking into account the meaning of these 225 genes selected

Author Response

(The authors gave the same response as above.)

Round 2

Reviewer 1 Report

The work has been substantially improved.  The discussion is now appropriately framed.  I appreciate the substantial editing of the manuscript. There remains a need to review the MS grammatically, as some sentences are difficult to follow. 

Reviewer 2 Report

Authors have adequately clarified all the issues raised. the manuscript is now publishable